# Equol: A Bacterial Metabolite from The Daidzein Isoflavone and Its Presumed Beneficial Health Effects

**DOI:** 10.3390/nu11092231

**Published:** 2019-09-16

**Authors:** Baltasar Mayo, Lucía Vázquez, Ana Belén Flórez

**Affiliations:** 1Departamento de Microbiología y Bioquímica, Instituto de Productos Lácteos de Asturias (IPLA), Consejo Superior de Investigaciones Científicas (CSIC), Paseo Río Linares s/n, 33300 Villaviciosa, Spain; lucia.vazquez@ipla.csic.es (L.V.); abflorez@ipla.csic.es (A.B.F.); 2Instituto de Investigación Sanitaria del Principado de Asturias (ISPA), Avenida de Roma s/n, 33011 Oviedo, Spain

**Keywords:** equol, daidzein, isoflavones, soy, soy products, gut metabolite, bioactive compound

## Abstract

Epidemiological data suggest that regular intake of isoflavones from soy reduces the incidence of estrogen-dependent and aging-associated disorders, such as menopause symptoms in women, osteoporosis, cardiovascular diseases and cancer. Equol, produced from daidzein, is the isoflavone-derived metabolite with the greatest estrogenic and antioxidant activity. Consequently, equol has been endorsed as having many beneficial effects on human health. The conversion of daidzein into equol takes place in the intestine via the action of reductase enzymes belonging to incompletely characterized members of the gut microbiota. While all animal species analyzed so far produce equol, only between one third and one half of human subjects (depending on the community) are able to do so, ostensibly those that harbor equol-producing microbes. Conceivably, these subjects might be the only ones who can fully benefit from soy or isoflavone consumption. This review summarizes current knowledge on the microorganisms involved in, the genetic background to, and the biochemical pathways of, equol biosynthesis. It also outlines the results of recent clinical trials and meta-analyses on the effects of equol on different areas of human health and discusses briefly its presumptive mode of action.

## 1. Introduction

Abundant epidemiological evidence suggests that diets rich in phytoestrogen-containing foods, such as soy and soy products, reduce the risk of a number of syndromes and chronic diseases, notably menopause symptoms in women, cardiovascular and neurodegenerative diseases, and certain types of cancer [1,2,3]. Isoflavones are non-nutritive phenolic compounds found in the roots and seeds of many plants, of these, soybeans are the richest source [4]. Soy isoflavones are phytoestrogens resembling 17-β-estradiol; although less active than the hormone, they show estrogen-like activity [5].

In plants, isoflavones are found mostly (>80%) in the form of glycoconjugates, i.e., the glucosides genistin, daidzin, and glycitin, and the corresponding acetyl and malonyl derivatives [6]. Glycosides are not readily absorbed in the gut and have only low-level estrogenic activity [7]. For isoflavones to become bioavailable and functional, these glycosides must be hydrolyzed into their corresponding isoflavone-aglycones, i.e., genistein, daidzein, and glycitein [8].

The amount of aglycones in plasma cannot be predicted from soy or isoflavone ingestion, as many intrinsic (genetic background, gut microbiota, bowel disease, age, sex, etc.) and extrinsic (isoflavone source, method of extraction, formulation, etc.) factors influence their bioavailability [9]. The plasma isoflavone concentration in humans is only 0.5–1.3% of that actually absorbed, much lower than in animal models (0.5–3.1% for rats and 3.1–26.0% in mice) [10]. Results obtained in animal studies may not, therefore, be easily extrapolated to humans. Absorbed aglycones are metabolized mainly to glucuronidated and sulphated derivatives by endogenous phase I and phase II enzymes [9,11]. They may then be further catabolized in the liver or secreted into the bile, thus returning to the intestine via the enterohepatic circulation [12].

Certain aglycone conjugates may have estrogenic activity too, and serve as an intracellular reservoir for the release of free aglycones in target cells [13]. However, as for absorption, extrapolation of plasma concentrations to the tissue level may not be valid. Indeed, the identification and quantification of the isoflavone derivatives in target tissues has rarely been determined [14]. In humans, equol concentrations have been reported to range between 22–36 and 456–559 nmol/kg in breast adipose and glandular tissue respectively [15,16].

Unabsorbed isoflavones and those excreted by the biliary system to the intestine reach the colon where they are deconjugated by bacterial enzymes and then (re)absorbed or further metabolized [12,17]. In the gut, isoflavone aglycones can be metabolized by intestinal microbes via several reactions, including reduction, methylation/demethylation, hydroxylation/dihydroxylation, and C-ring cleavage (Figure 1).

## 2. Equol

Extracted from the urine of pregnant mares back in 1932, equol [C_15_H_12_O(OH)_2_] was the first isoflavonoids to be identified [18]. Later, in 1982 it was the first isoflavone-derived compound detected in human urine and blood (reviewed by Setchell and Clerici [19]). Equol is an isoflavone-derived metabolite formed from daidzin/daidzein by bacteria in the distal region of the small intestine and colon [19]. From a chemical viewpoint, equol (4’,7-isoflavandiol) is an isoflavane phenolic compound with a non-planar structure, which might be responsible for its physiological activities [20]. Equol is optically active with an asymmetric carbon atom at the C3 position giving rise to *R*(-)- and *S*(-)-equol enantiomers. However, only *S*(-)-equol has been detected as a result of bacterial daidzein conversion [21,22,23]. Equol is more stable, more easily absorbed, and has a lower clearance than its precursor molecule daidzein [24]. It also shows stronger estrogenic activity than any other isoflavone or isoflavone-derived metabolite [23,25]. As for other isoflavones, equol also shows anti-androgenic activity by binding to and sequestering 5α-dihydrotestosterone [26]. In addition, it is the isoflavone-derived compound with the strongest antioxidant activity [27,28]; antioxidants are thought to have a prominent role in the onset and progress of different chronic diseases, including cancer [29].

### 2.1. Equol Production Phenotype

Equol is produced from the isoflavone daidzein in the gut of humans and animals by certain bacterial biotypes; those involved might differ between individuals [30]. All the animal species tested (mouse, rat, sheep, cow, goat, chicken, and fowl) have been shown to produce equol in response to soy or daidzein consumption [21]. However, it is produced by only 25–50% of human subjects (so-called equol producers); the percentage depends on the community in question and the dietary habits of its members [31,32,33,34]. In contrast, a majority of humans (80–90%) that do not produce equol (equol non-producers), convert a large part of daidzein into *O*-desmethylangolensin (*O*-DMA), a metabolite with no estrogenic activity [35,36]. *O*-DMA and equol are likely produced by different bacterial taxa.

Both observational and interventional studies have returned inconsistent results on the stability of the equol production phenotype in humans. Some authors propose equol production status to be rather stable [24,37], suggesting it to be under some degree of genetic control. However, other studies have reported the conversion, at small rates, of producers to non-producers and vice versa [38,39]. Thus, equol production appears to be stable in some but not all individuals. The frequency of equol producers among vegetarians has been reported significantly higher than among non-vegetarians (59% versus 25%) [34], suggesting that dietary components from plants other than soy itself may promote the ability to produce equol. Equol is not detected in the urine and plasma of most infants under one year of age [40,41], indicating that equol-producing bacteria are latecomers to the gastrointestinal ecosystem. The ability to produce equol may be influenced by shared environmental factors, although weak positive correlations between mothers and children have been reported [42,43].

Another topic of debate is whether dietary constituents other than isoflavones enhance equol formation in equol producers [38,44,45,46]. The consumption of resistant starch (together with daidzein) by ovariectomized mice has been correlated with enhanced equol excretion [47]. Similarly, the consumption of daidzein and lactulose has been reported to promote equol production in sows [48], and carbohydrate-rich diets have been shown to stimulate equol production in human fecal cultures [49]. The combined consumption of milk and dairy products with daidzein is also significantly correlated with equol excretion concentrations [50]. The antibiotic treatment of fecal cultures from different subjects has been shown to both increase and reduce equol production [46], again suggesting differences between people in terms of the equol-producing microorganisms carried. Together, these findings suggest that the equol production phenotype and equol production itself are modified by dietary habits.

### 2.2. Equol-Producing Microorganisms

It is well established that equol is formed from daidzein by gut bacteria [30,51]. However, our knowledge of the actual microorganisms involved is still limited [52]. Bacterial mixtures of different taxa capable of producing equol from daidzein have been described [49,53]. A few strains are known to convert daidzein into dihydrodaidzein although they cannot produce equol, while others can convert dihydrodaidzein into equol but do not act on daidzein (Table 1). In these cases, equol production has been detected by combining dihydrodaidzein producers (e.g., *Lactobacillus* sp. Niu-O16) with dihydrodaidzein utilizers (e.g., *Eggerthella* sp. Julong 732) [53]. In recent decades, a number of strains (from humans and animals) capable of forming equol have finally been identified (Table 1). Even, a few equol-producing bacteria can also metabolize genistein to generate 5-hydroxy equol [54,55,56], a compound highly similar to equol from a chemical point of view and with greater antioxidant activity than genistein [57].

Most of the equol-producing microbes isolated so far belong to the family Coriobacteriaceae [52]. Members of this family are also involved in the catabolism of cholesterol-derived compounds such as bile acids and corticoid hormones, hinting at their functional specialization in the gut [58]. The family Coriobacteriaceae includes genera such as *Adlercreutzia, Assacharobacter, Eggerthella, Enterorhabdus, Paraeggerthela*, and *Slackia* [52]. Among these, species such as *Adlercreutzia equolifaciens, Asaccharobacter celatus, Enterorhabdus mucosicola, Slackia isoflavoniconvertens*, and *Slackia equolifaciens* are reported to be equol producers (Table 1). Some strains have been identified only at the genus level; e.g., *Eggerthella* sp. YY7918, *Paraeggerthella* sp. SNR40-432, and *Slackia* sp. NATTS (Table 1). In spite of this, it is not yet sure whether equol production in the Coriobacteriaceae is a family-, species-, or strain-specific trait [59]. A few equol-producing strains of other taxa from either intestinal or food origin have recently been identified (Table 1), including *Bifidobacterium breve* ATCC 15700T, *Bifidobacterium longum* BB536, *Lactobacillus intestinalis* JCM 7548, *Lactobacilllus paracasei* CS2, *Lactobacillus sakei* CS3, *Lactococcus garvieae* 20-90, *Pediococcus pentosaceus* CS1, and *Proteus mirabilis* LH-52.

### 2.3. Molecular Aspects of Equol Formation

The bacterial biosynthesis of equol from daidzein proceeds via a series of consecutive reduction reactions (catalyzed by three reductases), involving the production of the intermediate compounds dihydrodaidzein and tetrahydrodaidzein (Figure 1). In *L. garvieae*, the genes involved in equol production are found in a 10 kbp operon-like structure [75,76,77]. At least three genes encoding a daidzein-dependent NADP reductase (*dzr*), a dihydrodaidzein reductase (*ddr*), and a tetrahydrodaidzein reductase (*tdr*), have been reported to be required for equol production in this bacterium. A fourth enzyme with dihydrodaidzein racemase activity, encoded immediately upstream of the reductase genes in the equol cluster, has been shown to be necessary for efficient equol production by *L. garvieae* [77]. All reactions in the pathway seem to be reversible [75]. More recently, next generation sequencing techniques have helped characterize the genomes of other equol-producing strains [78,79], which helped to revealed the genetic basis of the biochemical pathways involved in the synthesis of equol. In *S. isoflavoniconvertens*, the equol cluster is about 10.5 kbp in length and contains eight genes [80]. As in *L. garvieae*, the reductase enzymes in *S. isoflavoniconvertens* are encoded by homologous *dzr, ddr*, and *tdr* genes [80]. Equivalent genes encoding reductases similar to those of *L. garvieae* and *S. isoflavoniconvertens* have also been identified in *Slackia* sp. NATTS [81], *A. equolifaciens* [78], and *Eggerthella* sp. YY7918 [82]. Interestingly, the *L. garvieae* genes encoding the equol-related reductases have been reported similar to those in the Coriobacteriaceae (such as *S. isoflavoniconvertens, Eggerthella* sp., and *Slackia* sp. NATTS), arguing strongly for the recent horizontal transfer of the equol genetic makeup from a member of this family. This is further supported by the specific codon usage and high GC content (68%) of the equol-associated genes in *L. garvieae* [75,76], which greatly exceeds the genomic GC content of this species (39%). The genetic framework and the biochemical pathways of equol production in other non-Coriobacteriaceae species have yet to be determined.

The proteomic analysis of *S. isoflavoniconvertens* grown with daidzein has shown overexpression of the reductases and five other proteins encoded by genes located within the equol gene cluster [80]. Congruently, all enzymes of the cluster might be somehow involved in equol production and can be regulated in a coordinated manner. Indeed, the expression of 13 contiguous genes in the equol cluster of *A. equolifaciens* has recently been shown enhanced during the growth of this bacterium in the presence of daidzein, during which *dzr, ddr*, and *tdr* were the most strongly expressed genes [83]. Four expression patterns of transcription for the genes of the *A. equolifaciens* equol cluster were identified, although the operon seemed to be transcribed as a single RNA transcript [83]. The roles of other genes in the operon, (of which at least three encode flavoproteins that might be involved in oxidation-reduction reactions), and their regulatory mechanisms, are yet to be determined. In *S. isoflavoniconvertens*, the first enzyme of the pathway, daidzein reductase, has also been shown to participate in the transformation of genistein into dihydrogenistein, a key step in the formation of 5-hydroxy equol [80], a compound also produced by *E. mucosicola* [68].

The biotechnological synthesis of equol by anaerobic coriobacteria requires long culture times and is expensive. Cloning and expressing the genetic machinery of equol production in heterologous hosts might circumvent these challenges [80,81] while helping to reveal the function of all the determinants in the cluster. The genes encoding the three reductases of *L. garvieae* (*dzr, ddr*, and *tdr*) were soon cloned and expressed in *Escherichia coli* [75,76]. The same genes from *S. isoflavoniconvertens* and *Eggerthella* sp. YY7918 have been also cloned individually and expressed in *E. coli* [80,82]. This strategy identified the involvement of the *S. isoflavoniconvertens* reductases in the conversion of genistein into 5’-hydroxy-equol [80,84]. Expression of the genes in *E. coli* further allowed the development of recombinant strains with improved *S*(-)-equol production capacity [85,86]. A low production yield by fermentation has been reported when using recombinant microorganisms, perhaps due to the low solubility of isoflavones in aqueous systems. This problem has been recently overcome by adding hydrophilic polymers to the cultures [87]. Large scale production would surpass the current equol shortage, supplying enough for interventional studies that could assess its efficacy.

### 2.4. Equol-Producing Populations in the Human Gut

Since the gut microbiota plays an important role in the metabolism of soy isoflavones, understanding the role of soy and its components in influencing and modulating the gut microbiota is vital if we are to learn the mechanisms of action of soy’s bioactive compounds and promote their rational use in functional foods [88,89]. Equally important will be to know the types and numbers of microorganisms involved in the synthesis of equol present in the microbiota of different individuals. To that aim, based on conserved sequences of genes involved in equol synthesis from known equol-producing species, oligonucleotide primers have been developed to detect [90] and quantify [91] equol-producing microbes in samples of fecal origin. These primers have already been used to amplify reductase-encoding genes related to those of *A. equolifaciens* and *S. isoflavoniconvertens* in feces and fecal cultures [90,91]. However, no amplicons were obtained when DNA from fecal samples of certain equol producers was used as a template, suggesting the involvement of other yet unknown unrelated taxa in equol formation in these individuals [90,91]. Similar copy numbers of both *tdr* and *ddr* genes, about 4–5 log10 copies per gram of feces, have been reported [91]. As these are single-copy genes, an equal number of equol-producing bacteria must be expected. Indeed, these amounts are within the usual numbers for Coriobacteriaceae species in feces as quantified by 16S rRNA gene amplification [92,93,94]. Surprisingly, no significant increases in the copy number of equol-related genes (and thus bacteria) has ever been observed during isoflavone interventions [91] or after in vitro culturing of fecal samples, even under conditions in which equol production is favored (such as in carbohydrate-rich diets) (Vázquez et al., unpublished). These results imply that equol-producing bacteria are not positively selected for by isoflavones. It is also surprising that the fact that genes involved in equol production and equol-producing bacterial numbers have been reported in both equol-producing and non-producing individuals [44,91]. Altogether these results suggest that more studies will be required if we are to understand the composition and changes in equol-producing populations in the gut. As minority populations, deciphering the interaction of the equol producing microbes with majority and pivotal microbial communities within the gut ecosystem is paramount.

## 3. Soy, Soy Isoflavones, Equol, and Health

In East Asian countries, climacteric vasomotor symptoms during menopause in women are less severe than in Western women and the incidence of cardiovascular disease, osteoporosis, mental disorders, and certain types of cancer is about two- to four-fold of that seen in the West [1,2,3,95,96,97]. Alongside genetic factors, this large difference is assumed to have a nutritional basis. Isoflavones are an important component of Asian diets (15–50 mg day versus <2 mg in Western countries) [98,99,100,101], and observational and epidemiological studies have correlated a high intake of soy and isoflavones with reduced menopause symptoms, increased bone formation, reduced bone resorption, improved learning, and a reduced risk of prostate, colon, and breast cancer [102,103,104,105].

In vitro laboratory studies and animal interventions can predict the impact of isoflavones on human health, but only human trials can provide proof. However, most current human interventions involving isoflavones have suffered from small sample sizes, short trial durations, lack of appropriate controls, the use of isoflavones from various sources, supplements with different aglycone contents, and other methodological flaws [106]. Not surprisingly, this has led to inconsistent results being reported [96,107,108,109,110,111]. Indeed, most reviews and meta-analyses report the results of soy and isoflavone intervention studies to be far from conclusive [1,112,113,114,115,116,117]. As a result, regulatory agencies usually conclude there to be no scientifically sound evidence of isoflavones reducing the risks and symptoms of any disease [14,106,118]. In addition to the effect of genetic variation on the phenotypic expression of human disease [119,120], interpersonal differences in the intestinal microbiota may also account (at least in part) for the discrepancies seen [121,122]. Such differences could give rise to different microbial isoflavone-derived metabolites being produced [12,33], which might explain the lack of effectiveness in some studies. In particular, there has been much speculation regarding the reason why just a fraction (25 to 50%) of the human population produces equol. Conceivably, these subjects might be the only ones who would benefit from soy or isoflavone consumption [111,123,124]. To test this hypothesis, the categorization of the individuals in isoflavone trials by their equol-producing phenotype is pivotal. This only began in recent years [32,96,124,125,126] and no firm conclusions have yet been drawn. Indeed, the results of many studies have been very conflicting [108,109,110,111,127,128,129,130].

In contrast to their possible health benefits, the anti-estrogenic properties of isoflavones might also cause them to act as unwanted endocrine disruptors [131]. In vitro and animal studies both report isoflavones able to interfere with different checkpoints of the hypothalamic/pituitary/thyroid system [132]. This could have a huge repercussion on thyroid homeostasis. Further, the estrogenic activity of isoflavones (and thus equol) could pose a potential hazard by promoting certain types of tumor [133]. However, the scientific evidence supporting their having any harmful consequences is also inconclusive [14,106].

### 3.1. Equol, Menopause, and the Cardiovascular System

Soy intake has been correlated with fewer hot flushes and night sweats during menopause [1,134]. In addition, there is growing evidence that soy and soy isoflavones may help regulate vasoactivity [135], as well as lipid metabolism and cholesterol transport [136,137,138].

Equol offers an alternative for the management of menopausal symptoms, extending the otherwise reduced benefits of soy isoflavones or isoflavone supplement consumption to beyond equol producing women. Intervention trials in humans have frequently focused on non-equol-producing populations [107,139,140,141]. Compared to placebo controls, beneficial effects have been reported for women taking an *S*(-)-equol supplement (10 to 30 mg daily for 8–12 weeks) with respect to the major menopausal symptoms (hot flushes) [139,142] and arterial stiffness [143]. A recent meta-analysis also revealed a significant reduction in hot flash scores (incidence and/or severity) following equol supplementation in both equol-producing and non-producing menopausal women [144]. Thus, equol might serve as a new, promising, and safe therapeutic option to be used as complementary therapy for women with vasomotor symptoms.

In the Orient, age-adjusted mortality rates of cardiovascular diseases are lower of that seen in Western countries and inversely correlated to isoflavone excretion in urine [98,134]. Recent observational studies and short-term randomized controlled trials have correlated equol with a reduced risk of coronary heart disease via its enhanced anti-atherogenic potential and the improvement of arterial stiffness [145]. Interventions with natural equol in overweight Japanese men and women suggested it may help in the prevention of cardiovascular diseases by lowering low-density lipoprotein cholesterol (LDL-C) levels and improving the cardio-ankle vascular index (a blood pressure-independent index of arterial stiffness) [146]. However, no vascular benefits (arterial stiffness, blood pressure, endothelial function, and nitric oxide formation) were observed in equol non-producing men after the acute intake of equol (40 mg) [107].

### 3.2. Equol and Bone Health

Isoflavones have been repeatedly reported to help prevent osteoporosis, a major problem for menopausal women [104,130,147,148]. The exact mechanism by which isoflavones preserve bone health is not completely understood. It seems that isoflavones trigger the activity and proliferation of osteoblasts via insulin-like growth factor 1 (IGF-1), a key factor in maintaining bone mass against the action of osteoclasts [149]. In a meta-analysis study, Wei and coworkers have found that, as compared to baseline levels, intake of soy isoflavone supplements significantly increased bone mineral density and decreased the bone resorption marker urinary deoxypyridinoline [104]. However, in the same study, no significant effect on serum bone alkaline phosphatase activity (which is involved in bone formation) was observed. In mice, equol has been shown to reduce the expression of genes associated with the inhibition of bone formation, osteoclast and immature osteoblast specificity, and cartilage destruction [150]. In humans, the treatment of postmenopausal women with 10 mg/day of equol for one year prevented a reduction in bone mineral density in the entire body [143]. This work further showed that equol supplementation markedly inhibited bone resorption, as demonstrated by reduced urinary deoxypyridinoline excretion concentrations.

### 3.3. Equol and Cancer

The incidence of prostate, colon, and breast cancers is much lower in East Asian countries than in the West [2,3,93,96,151,152]. Although environmental factors are thought to contribute strongly to the development of tumors, Asian immigrants to Western countries who change their dietary habits suffer from these forms of cancer at similar rates to Westerners, which suggests that isoflavones, via soy consumption, might be related to this reduction in risk [96]. Further, recent evidence suggests there is a reduced risk of developing breast cancer if soy was consumed during childhood and/or adolescence [153].

Only a few observational studies have investigated the influence of equol and the equol-producer status on breast cancer incidence or markers of breast cancer risk. Certainly, the plasma concentration of equol in women with breast cancer has been found similar to that of healthy controls [154], and no association between equol-producer status and breast cancer risk has ever been established [155]. Indeed, contradictory results have also been obtained, even from the same human group (the EPIC-Norfolk Cohort) [10,156]. Controversial results have also been reported regarding breast density in women (another marker of breast cancer) and soy food or soy supplement consumption, and the equol producer or non-producer phenotype [155,157].

In men, the ability to produce equol or equol itself has been suggested to help in reducing the incidence of prostate cancer [39]. The results also suggest that a diet based on soybean isoflavones could be useful in preventing prostate cancer.

### 3.4. Equol and the Central Nervous System

Epidemiological studies reveal lower rates of dementia in East Asian populations [158]. Studies on the effects of isoflavones on the gut-brain axis in humans have focused mostly on cognitive functions. In general, beneficial effects have been reported [159,160,161,162,163]. The long-term administration of soy or extracted isoflavones has been associated with improved learning, logical thinking, and planning ability in menopausal women. However, inconclusive findings on the neuroprotective effects of isoflavones and other phytoestrogens have also been reported [164]. Evidence is, however, accumulating that atherosclerosis and arterial stiffness are positively associated with cognitive decline. Sekikawa et al. [145], who showed that *S*-equol was anti-atherogenic and could improve arterial stiffness, reported in their review that equol may help prevent cognitive impairment/dementia.

### 3.5. Equol and Other Health Benefits

A topical equol intervention suggested equol to have an anti-aging effect on the skin of postmenopausal women, reducing wrinkle area and depth [165]. More recently, equol application to the skin has also been associated with an improvement in skin roughness, texture, and smoothness, and in some epigenetic molecular markers (LINE-1 methylation and telomere length) in skin cells [166]. Not surprisingly, the compound has recently attracted substantial attention of the cosmetic industry.

Equol has also been suggested to modulate obesity and diabetes type-2 by controlling the glycemic index [146,167], and to ameliorate chronic kidney disease [103].

## 4. Mechanistic Mode of Action of Equol

The mechanistic mode of action of equol is not yet completely understood. Studies have mostly been done through in vitro assays using concentrations higher than those found under physiological conditions, thus limiting the provision of robust and definitive conclusions. Further, equol is found in plasma mainly as a 7-*O*-glucuronide derivative [168], which makes it difficult to discern the biologically-active form(s) at tissue and cellular levels. In spite of these deficits, evidence from experimental studies suggests that equol may act in multiple ways [169]. Based on its structural similarity to 17-β-estradiol, equol binds to both estrogen receptors (ERs) α (ERα) and β (ERβ—the preferred target) with greater affinity than its precursor daidzein, and to a degree comparable to that of genistein [170,171]. As ERs are not equally distributed among the different tissues, equol might have different effects depending on the ratio of ERα and ERβ isoforms present. Whether it acts as an agonist or an antagonist may further depend on the level of endogenous estrogens present, as they bind to both receptors more tightly [172]. The antioxidant activity of equol seems to be mostly mediated by its interaction with the ERβ [27], which induces the extracellular signal-regulated protein kinases (ERK1/2) and the NF-κB peptide, factors that control transcription, cytokine production, and cell survival [173]. Isoflavones and equol may not act as antioxidants themselves but rather by triggering cell signaling pathways leading to changes in the expression of cellular enzymes such as superoxide dismutase, catalase, and glutathione peroxidase (all involved in counteracting oxidative stress) [29]. Mechanistically, equol’s influence on transcription seems to proceed through the activation of the transactivation function AF-1 [174]. Another mode of action of equol underlying several physiological effects may relate to epigenetic mechanisms, including DNA methylation, histone modification, and microRNA regulation [175].

Equol has been reported to induce acute endothelium- and nitric oxide (NO)-dependent relaxation of the aortic rings, and is a potent activator of the human and mouse pregnane X receptor (PXR), a steroid and xenobiotic sensing protein in the nucleus [176]. Further, it has been proposed that it modulates endothelial redox signaling and NO release, involving the transactivation of the epidermal growth factor receptor kinase (EGFRK) and the reorganization of the F-actin cytoskeleton [177]. Equol has also been shown to prevent (at physiological concentrations) oxidized LDL-stimulated apoptosis in human umbilical vein endothelial cells [178], and to reduce the oxidative stress induced by lipopolysaccharides in chicken macrophages [179]. These activities may provide the basis for therapeutic strategies, for instance by restoring endothelial function in cardiovascular diseases. An improvement in atherosclerosis has also been reported via equol attenuating ER stress, mediated by the activation of the NF-E2 p45-related factor 2 (Nrf2) signaling pathway [180]. The cancer-protective effects of isoflavones and equol have been attributed to a variety of signaling pathways, including the regulation of the cell cycle (by reducing the activity of the cyclin B/CDK complex) [181], the inhibition of cell proliferation (by, for instance, reducing activity of topoisomerase II) [182], the induction of apoptosis [173,183,184], and the degradation of androgen receptor by S-phase kinase-associated protein 2 (PKAP2) [185]. The anti-prostate cancer activity of equol in cell cultures has been proposed associated with activation of FOXO3a (one of the forkhead-family factors of transcription involved in apoptosis) via protein kinase B (Akt)-specific signaling transduction pathway, and with the inhibition of the expression of the MDM2 complex (a negative regulator of tumor suppressor p53) [186,187], plus the inhibition of the degradation of the androgen receptor [185]. Diabetes and other metabolic disorders may be influenced by equol via its preventing glucagon-like peptide 1 (GLP-1) secretion from the GLUTag cells [188]. The modulation of glucose-induced insulin secretion and the suppression of glucagon release (from the α- and β-pancreatic cells, respectively) by GLP-1 in response to the ingestion of nutrients have been firmly established [189]. Equol can also prevent hypoglycemia by activating cAMP signaling at the plasma membrane of INS-1 pancreatic β-cells [190]. Finally, it has been reported to significantly increase the expression of genes coding for collagen, elastin, and tissue inhibitor of metalloproteases, while reducing the expression of metalloproteinases [191]. All these factors contribute towards an improvement of the skin’s antioxidant status, delaying aging. However, despite of all the knowledge gathered by these in vitro observations, the effects of equol on human health in vivo, and their magnitude, are yet to be confirmed.

## 5. Conclusions

In summary, current interest in dietary isoflavones has been driven by epidemiological studies, suggesting that diets rich in phytoestrogens are beneficial to human health. Soy isoflavones and isoflavone-derived metabolites are structurally similar to estrogen and might have some of its effects. Equol is a key isoflavone-derived metabolite with estrogenic and antioxidant activities. Studies examining the influence of equol and the equol-producer status on several disease conditions have been inconclusive, and there is an urgent need for large-scale, well designed, randomized, double blind, placebo-controlled human trials. Further knowledge is also required on the changes in metabolic markers induced by isoflavones/equol interventions. Understanding the gut microbial populations involved in equol biosynthesis, and their regulatory mechanisms, is also pivotal for maximizing endogenous equol production. Identification of the actual compound(s) with a role in the signaling cascades underlying the involved cellular and physiological processes would contribute greatly to the functional characterization of the role of this bioactive metabolite. The exploitation of equol-producing microorganisms or their genetic machinery for the biotechnological production of this bioactive agent would allow the use of equol in large-scale interventional trials. This would ultimately serve to test the real involvement of equol in human health. Finally, well characterized equol-producing strains could be used in the future as probiotics for animals and humans aimed as a means of increasing equol production in the gut.

## Figures and Tables

**Figure 1 nutrients-11-02231-f001:**
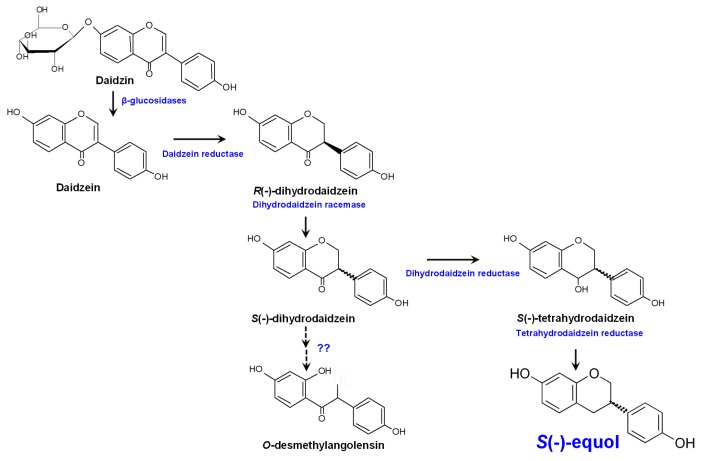
Metabolism of the isoflavone glucoside daidzein by the human gut microbiota and equol biosynthesis pathway.

**Table 1 nutrients-11-02231-t001:** Bacterial species and strains involved in the metabolism of equol or its intermediate precursors from daidzein.

Species	Strain/s	Origin	Reference
*Adlercreutzia equolifaciens*	FJC-B9 ^T^	Human feces	Maruo et al. [60]
* Asaccharobacter celatus *	do03 ^T^	Rat cecum	Minamida et al. [61]
* Bifidobacterium breve *	ATCC15700 ^T^	Human intestine	Elghali et al. [62]
* Bifidobacterium longum *	BB536	Human feces	Elghali et al. [62]
* Catenibacterium * sp.	D1	Human feces	Yu et al. [63]
*Clostridium* sp.	HGH6 ^a^	Human feces	Hur et al. [64]
*Clostridium*-like sp.	TM-40 ^a^	Human feces	Tamura et al. [65]
*Eggerthella* sp.	YY7918	Human feces	Yokoyama and Suzuki [66]
*Eggerthella* sp.	D2	Human feces	Yu et al. [63]
*Eggerthella* sp.	Julong 732 ^b^	Human feces	Wang et al. [67]
*Eggerthella*-like bacteria	SNR48-44, SNR44-10, SNR45-571, SNR46-41, SNR48-350	Stinky tofu	Abiru et al. [54]
* Enterorhabdus mucosicola *	Mt1B8 ^T^	Mouse ileal mucosa	Matthies et al. [68]
* Lactobacillus * sp.	Niu-O16 ^a^	Bovine rumen	Wang et al. [67]
* Lactobacillus paracasei *	CS2 (JS1)	Human feces	Kwon et al. [69]
* Lactobacillus sakei/graminis *	CS3	Human feces	Kwon et al. [69]
* Lactobacillus intestinalis *	JCM 7548	Rat feces	Heng et al. [70]
* Lactococcus garvieae *	20-92	Human feces	Uchiyama et al. [71]
* Paraeggerthella * sp.	SNR40-432	Stinky tofu	Abiru et al. [54]
* Pediococcus pentosaceus *	CS1	Human feces	Kwon et al. [69]
* Proteus mirabilis *	LH-52	Rat intestine	Guo et al. [72]
* Slackia equolifaciens *	DZE ^Tc^	Human feces	Jin et al. [73]
* Slackia isoflavoniconvertens *	HE8 ^Tc^	Human feces	Matthies et al. [55]
* Slackia * sp.	NATTS	Human feces	Tsuji et al. [74]

^a^ Daidzein to dihydrodaidzein only. ^b^ Equol from dihydrodaidzein only. ^c^ These strains are also able to produce 5-hydroxy equol from the isoflavone genistein. The T superscript denotes the isolate as the species type strain.

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
