# Peer review of "Equol: A Bacterial Metabolite from The Daidzein Isoflavone and Its Presumed Beneficial Health Effects"

_nutrients, 2019, doi:10.3390/nu11092231_

Round 1

Reviewer 1 Report

The review is clearly written and well organized.

Line 2: does not sound to me “a bacterial a metabolite” better a bacterial metabolite.

Line 62: delete –

Line 199-201: please re-write the sentence, it  is not clear enough

Line 213: is also surprisingly that… instead of Also surprising is the fact 

Reviewer 2 Report

This is an interesting review of equol with extensive referencing and, for the most part, a balanced perspective. There are some suggestions and corrections.

Line 201.  "quantification" is not the best word.  Should use "quantify". Lines 221 - 224.  It is not possible that the incidence of CVD and other disorders is 1/8 seen in the West.  It is plausible that there is a delay in CVD, for example, but CVD and cancer are still responsible for the majority of the deaths. Line 236.  The reference should be #106, not #105. Lines 269-270.  Again, the statement that in the Orient that CVD "incidence" is 12-13% of the West seems implausible.  Perhaps one needs to define "incidence". Lines 283-285.  It is also not plausible that any dietary intervention would increase bone mineral density by 50%.  Drugs are not able to do this.  Maintaining bone density or few percentage increase over time are the best outcomes to be expected. Lines 327 -372.  This section on "Mechanisms" needs some perspective.  These are largely in vitro studies using very high molar concentrations of isoflavones compared to physiological quantities.  The authors should be careful to point this out either by each study or overall. Line 358.  There is a typo for reference #182. Line 377.  Use of the word "powerful" is not justified as there is little to no in vivo evidence for this. Line 650.  Typo "atherosclerosis" 

Round 2

Reviewer 2 Report

The authors have carefully addressed the concerns of the reviewer and have further done additional editing that has enhanced the manuscript.  The only point to me made is that in the References section all references are numbered twice (in other words, for example reference 4 reads "4.  4. Aguiar......" - which seems unusual.